# Trainable Nonexpansive Denoisers for Contractive Image Reconstruction

**Arghya Sinha** [1]   **Aditya Banerjee** [1]   **Trishit Mukherjee** [1]   **Kunal N. Chaudhury** [1]

## Abstract

Trainable denoisers with Lipschitz control have become central to convergent image reconstruction. However, training neural networks that simultaneously offer strong denoising performance and global Lipschitz guarantees is challenging. Existing approaches enforce Lipschitz control only empirically, providing no guarantees beyond the training data. In this work, we show that by exploiting the action of permutations on the image lattice, we can constrain a neural architecture that is globally nonexpansive (Lipschitz bound $\leqslant 1$). We integrate the proposed denoiser with forward imaging operators to develop a reconstruction mechanism that is provably contractive and therefore globally convergent. Experiments on standard inverse problems, such as superresolution and deblurring, demonstrate that our reconstruction performance is competitive with softly constrained baselines while providing Lipschitz guarantees.

## 1. Introduction

Recovering an image from noisy and incomplete linear measurements is a fundamental problem in computational imaging, with applications in deblurring, superresolution, magnetic resonance imaging, etc. (Elad et al., 2023; Bouman, 2022). The measurement process is commonly modeled as

$$\boldsymbol{y} = A\bar{\boldsymbol{x}} + \boldsymbol{\epsilon}, \qquad (1)$$

where $\boldsymbol{y} \in \mathbb{R}^m$ denotes the observed data, $A \in \mathbb{R}^{m \times n}$ is a known forward operator, $\bar{\boldsymbol{x}} \in \mathbb{R}^n$ is the unknown image, and $\boldsymbol{\epsilon}$ represents measurement noise. Since (1) is typically ill-posed, recovering $\bar{\boldsymbol{x}}$ requires incorporating prior information about the image. A classical approach formulates image reconstruction as a variational problem

$$\min_{\boldsymbol{x} \in \mathbb{R}^n} \; f(\boldsymbol{x}) + g(\boldsymbol{x}), \quad f(\boldsymbol{x}) = \tfrac{1}{2}\|A\boldsymbol{x} - \boldsymbol{y}\|^2, \qquad (2)$$

where $f$ is a data-fidelity term and $g$ is a regularizer encoding prior knowledge (Bouman, 2022). Such problems are commonly solved using proximal algorithms, including proximal gradient descent (PGD), half-quadratic splitting (HQS), and ADMM (Bauschke & Combettes, 2011). In practice, the reconstruction quality depends strongly on the choice of regularizer $g$, which is often difficult to design.

**Implicit regularization.** Instead of handcrafting an explicit regularizer, plug-and-play (PnP) methods replace the proximal operator with a general denoising map $\mathcal{D} : \mathbb{R}^n \to \mathbb{R}^n$. For example, the PnP-HQS update (with $\rho > 0$) takes the form

$$\boldsymbol{x}_{k+1} = \mathcal{D}\big(\mathrm{prox}_{\rho f}(\boldsymbol{x}_k)\big) \qquad \text{(PnP-HQS)}, \qquad (3)$$

where the denoiser plays the role of an *implicit* regularizer, avoiding the need to design $g$ explicitly. This popular idea was introduced in (Venkatakrishnan et al., 2013) and has since been used with a variety of base algorithms, including PnP-PGD and PnP-ADMM (Hurault et al., 2022b; Wei et al., 2024; 2025). A fundamental advantage of PnP is that decoupling the denoiser from the forward model makes training independent of the forward operator $A$ and the downstream task, unlike end-to-end approaches, where the training data is tied to a specific application. As a result, many pretrained deep denoisers have been successfully reused across a wide range of inverse problems.

PnP uses the denoiser *iteratively*, repeatedly applying the same network within a fixed-point algorithm. This repeated use is outside the setting in which the network is typically trained for, and it often leads to unstable behavior. Consequently, PnP methods generally lack inherent convergence guarantees when $\mathcal{D}$ is treated as a black-box denoiser.

This observation has motivated extensive research on identifying structural conditions under which PnP algorithms converge. Such approaches for PnP convergence can be divided into two categories. The first category seeks to associate the denoiser with an explicit potential function. In (Sreehari et al., 2016; Moreau, 1965), it was shown that if the Jacobian $\nabla \mathcal{D}(x)$ is symmetric with eigenvalues in $[0, 1]$, then $\mathcal{D}$ can be interpreted as the proximal operator of a proper, closed, and convex function. However, this condition is violated by most practical denoisers such as DnCNN (Zhang et al., 2017a), DRUNet (Zhang et al., 2021),

---

[1]Indian Institute of Science, Bengaluru, India. Correspondence to: Arghya Sinha <arghyasinha@iisc.ac.in>.

*Proceedings of the 43rd International Conference on Machine Learning*, Seoul, South Korea. PMLR 306, 2026. Copyright 2026 by the author(s).

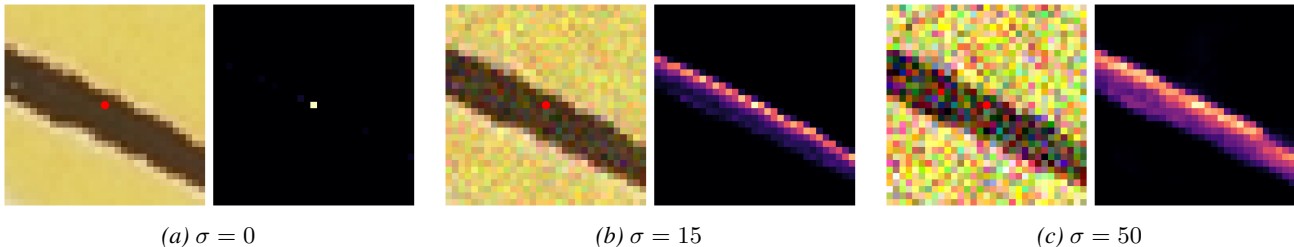

*(a) $\sigma = 0$*        *(b) $\sigma = 15$*        *(c) $\sigma = 50$*

*Figure 1.* **What does the network $\mathcal{N}_\theta$ learn?** Once trained, the network learns to highlight regions that are truly similar. Each panel shows two images: the noisy image with noise level $\sigma$ on the left, and a colormap on the right. The colormap shows the contribution, determined by $\mathcal{N}_\theta$, of all pixels reached by the permutations to denoising the central red pixel. When the noise is zero, the contributions from all pixels other than the red pixel are zero. As the noise increases, the neural network $\mathcal{N}_\theta$ learns to assign weight only to regions that are similar to the red pixel. When there is no noise, the network assigns small weights to mismatches, effectively avoiding aggregation that would introduce smoothing. As $\sigma$ increases, the network assigns more weight to similar regions, which helps suppress noise through aggregation. The learned separation is sharp, helping prevent diffusion and avoid unnecessary blurring.

SCUNet (Zhang et al., 2023) etc. Regularization by denoising (RED) (Romano et al., 2017) relaxes this requirement by defining an explicit regularizer from $\mathcal{D}$, but its convergence guarantees rely on assumptions such as local homogeneity and Jacobian symmetry, which are known to fail for deep denoisers (Reehorst & Schniter, 2018). More recent work parameterizes the potential function directly using neural networks (Cohen et al., 2021), or learns gradient-step and proximal-style denoisers (Hurault et al., 2022a;b).

We are primarily interested in the second line of work, where PnP algorithms are modeled as nonlinear dynamical systems (Ryu et al., 2019; Gavaskar et al., 2021; Zhang et al., 2021; Kawar et al., 2022) and their convergence is studied through fixed-point theory (Bauschke & Combettes, 2011). For example, (3) defines the following fixed-point iteration with the fixed-point (reconstruction) operator $\mathcal{T}$:

$$\boldsymbol{x}_{k+1} = \mathcal{T}(\boldsymbol{x}_k), \quad \mathcal{T} = \mathcal{D} \circ \mathrm{prox}_{\rho f}, \qquad (4)$$

whose convergence depends critically on the Lipschitz properties of $\mathcal{D}$ (Bauschke & Combettes, 2011). Within this framework, convergence guarantees can be obtained either by using classical denoisers with well-understood operator properties (Sreehari et al., 2016; Rudin et al., 1992), or by learning deep denoisers that are designed to satisfy Lipschitz-type control conditions (Pesquet et al., 2021; Hertrich et al., 2021; Goujon et al., 2023; 2024; Ducotterd et al., 2024). A central obstacle is that estimating the exact Lipschitz constant of a neural network is computationally intractable in general (Virmaux & Scaman, 2018), and enforcing global Lipschitz constraints is often restrictive and may reduce denoising performance. In practice, the key challenge is to maintain the expressiveness of the denoiser while imposing the structural conditions required for convergence.

**Motivation.** Recent works (Pesquet et al., 2021; Wei et al., 2024; 2025) take a reasonable approach to this trade-off. Since imposing global Lipschitz constraints directly

on high-performing unconstrained architectures, such as DRUNet (Zhang et al., 2021) and DnCNN (Zhang et al., 2017b), can significantly degrade performance, they enforce the required Lipschitz control only on finite training samples via penalty terms, rather than globally over the entire domain. This strategy significantly improves reconstruction quality relative to globally constrained baselines. However, because the constraint is enforced only empirically on the training distribution, it does not yield guarantees outside the training set, and the learned denoisers remain unconstrained on the rest of the domain. This raises a fundamental question: to what extent can one impose *global* constraints on denoisers while retaining sufficient expressive power for high-quality reconstruction? Motivated by this question, we pursue the following design goals:

- Avoid estimating and constraining the Lipschitz constant of the network. Also avoid soft-enforcement via sample-based penalty during training.

- Construct a class of globally Lipschitz operators that can be trained end-to-end to yield a strong denoiser.

- Using this as an implicit regularizer, develop a convergent reconstruction framework.

To this end, we want to develop a nonexpansive denoiser $\mathcal{D}$ (see Section 2) satisfying

$$\|\mathcal{D}(\boldsymbol{x}) - \mathcal{D}(\boldsymbol{y})\|_2 \leqslant \|\boldsymbol{x} - \boldsymbol{y}\|_2, \qquad (5)$$

for all $\boldsymbol{x}, \boldsymbol{y}$, i.e., $\mathcal{D}$ has Lipschitz constant at most 1. Enforcing such a global constraint directly on a highly nonlinear network is difficult since its Jacobian varies with the input. Purely linear denoisers are easier to control but tend to be too weak in practice (see Table 1). Our approach is to combine both: we use nonlinearity only to predict weights, and apply a linear map to the image itself. We achieve this through *weighted aggregation* (see Section 2).

Many classical image restoration methods use weight aggregation of neighboring pixels with data-dependent

weights (Buades et al., 2005; Dabov et al., 2007; Arias-Castro et al., 2012). Later works such as (Wang et al., 2018) extend this weighting principle to learned feature spaces. Formally, some nonlinear operators project the inputs to different feature spaces, and the weights determine which features to prioritize. This is a very practical and general notion of weighted aggregation, and many works have used it within neural network architectures to capture long-range dependencies and align correlated structures (Wang et al., 2018; Xia et al., 2020; Zhang et al., 2019). In fact, attention mechanisms (Vaswani et al., 2017) can be viewed as a particular instance of this type of aggregation (Wang et al., 2018).

In our case, we keep the nonlinearity only in the *weight prediction*, and make the denoising step an *interpretable linear aggregation*. We formalize this construction in Section 2 and use it in Section 3 to establish our main guarantees (see Lemma 3.3) under mild assumptions.

**Contribution.** The main contributions of this paper are:

- We propose a principled framework for constructing a parametric family of denoisers that are *nonexpansive by design* for all parameter values.

- We prove that the reconstruction operator $\mathcal{T}$ (in (4)) with the proposed denoisers is *contractive* under mild conditions (Theorem 3.1).

- We parameterize the denoiser using a lightweight CNN architecture, and demonstrate empirically that the proposed approach achieves competitive reconstruction quality on standard inverse problems while providing certified stability and convergence guarantees.

**Organization.** We present the denoiser and reconstruction algorithm in Section 2, prove nonexpansivity and contractivity in Section 3, and report the implementation details and experimental results in Sections 4 and 5.

## 2. Nonexpansive Denoiser

To construct the denoiser, we first generate different variants of the same image by applying permutations to its pixel indices. This is naturally described in terms of permutation group actions on the image lattice. The following abstraction is purely technical and is needed to establish the convergence analysis in Theorem 3.1. The practical implications are discussed in Section 4. Let $\Omega \subset \mathbb{Z}^2$ be the lattice of pixel indices, and represent an image as a function $\boldsymbol{x} : \Omega \to \mathbb{R}$, or equivalently as a vector in $\mathbb{R}^{|\Omega|}$. We denote the space of all images defined on $\Omega$ by $\mathcal{X} := \mathbb{R}^{|\Omega|}$.

Now let $\mathcal{G}$ be a group with identity element $e$ acting on an arbitrary set $X$. A (left) group action of $\mathcal{G}$ on $X$ is a

mapping $(g, x) \mapsto g \cdot x$ from $\mathcal{G} \times X$ to $X$ satisfying the axioms (Dummit & Foote, 2004),

$$e \cdot x = x, \qquad (g_1 g_2) \cdot x = g_1 \cdot (g_2 \cdot x), \qquad (6)$$

for all $g_1, g_2 \in \mathcal{G}$ and $x \in X$. We now specialize this definition to image data by letting $\mathcal{G}$ be a group of permutations of the image lattice $\Omega$, i.e., each element $\pi \in \mathcal{G}$ is a bijection $\pi : \Omega \to \Omega$, and the group operation is composition.

The action of a permutation $\pi \in \mathcal{G}$ on an image $\boldsymbol{x} \in \mathcal{X}$ is defined by the pullback

$$(\pi \cdot \boldsymbol{x})(i) = \boldsymbol{x}(\pi^{-1} i) \qquad (i \in \Omega). \qquad (7)$$

This definition ensures that the group action axioms are satisfied, and thus induces a valid left group action on the image space $\mathcal{X}$.

We now define the proposed denoiser $\mathcal{D}$. Let $\mathcal{N}_\theta : \mathcal{X} \times \mathcal{X} \to \mathcal{X}$ denote a neural network with trainable parameters $\theta$, and let $\boldsymbol{x} \in \mathcal{X}$ be a noisy image. We define the unnormalized aggregation operator $\mathcal{K} : \mathcal{X} \to \mathcal{X}$ as

$$\mathcal{K}(\boldsymbol{x}) = \sum_{\pi \in \mathcal{G}} \mathcal{N}_\theta(\boldsymbol{x}, \pi \cdot \boldsymbol{x}) \odot (\pi \cdot \boldsymbol{x}), \qquad (8)$$

where $\odot$ denotes element-wise multiplication. The denoiser $\mathcal{D}$ is obtained by applying element-wise normalization:

$$\mathcal{D}(\boldsymbol{x}) = \mathcal{K}(\boldsymbol{x}) \oslash C(\boldsymbol{x}), \qquad (9)$$

with

$$C(\boldsymbol{x}) = \sum_{\pi \in \mathcal{G}} \mathcal{N}_\theta(\boldsymbol{x}, \pi \cdot \boldsymbol{x}), \qquad (10)$$

where $\oslash$ is the element-wise division. For the division to be valid, we need $C(\cdot) > 0$. *We ensure this by imposing $\mathcal{N}_\theta(\boldsymbol{x}, \boldsymbol{y}) > 0$ for all $\boldsymbol{x}, \boldsymbol{y} \in \mathcal{X}$ through a trainable positive activation* (A2).

We can decouple the image passed through the neural network from the image being aggregated. Specifically, let $\boldsymbol{x}, \boldsymbol{\xi} \in \mathcal{X}$. We define the aggregation operator $\mathcal{K} : \mathcal{X} \times \mathcal{X} \to \mathcal{X}$ as

$$\mathcal{K}(\boldsymbol{x}; \boldsymbol{\xi}) = \sum_{\pi \in \mathcal{G}} \mathcal{N}_\theta(\boldsymbol{\xi}, \pi \cdot \boldsymbol{\xi}) \odot (\pi \cdot \boldsymbol{x}), \qquad (11)$$

where $\boldsymbol{\xi}$ is the reference image. The corresponding denoiser is obtained by normalizing the aggregate, following (9):

$$\mathcal{D}(\boldsymbol{x}; \boldsymbol{\xi}) = \mathcal{K}(\boldsymbol{x}; \boldsymbol{\xi}) \oslash C(\boldsymbol{\xi}), C(\boldsymbol{\xi}) = \sum_{\pi \in \mathcal{G}} \mathcal{N}_\theta(\boldsymbol{\xi}, \pi \cdot \boldsymbol{\xi}) \quad (12)$$

This formulation separates the roles of $\boldsymbol{x}$ and $\boldsymbol{\xi}$. The input $\boldsymbol{x}$ only appears inside the linear aggregation, while the reference $\boldsymbol{\xi}$ determines the adaptive weights through $\mathcal{N}_\theta$. As a result, for any fixed $\boldsymbol{\xi}$, the operators $\mathcal{K}(\cdot; \boldsymbol{\xi})$ and $\mathcal{D}(\cdot; \boldsymbol{\xi})$

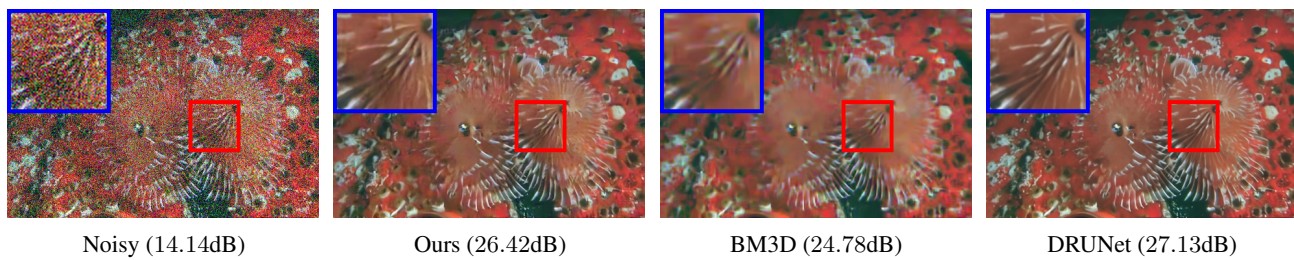

Noisy (14.14dB)      Ours (26.42dB)      BM3D (24.78dB)      DRUNet (27.13dB)

*Figure 2.* Denoising at noise $\sigma = 50/255$.

are linear in $\boldsymbol{x}$, while remaining nonlinear in $\boldsymbol{\xi}$. We will use this structure in two ways: the nonlinearity in $\boldsymbol{\xi}$ provides expressive, data-adaptive weighting, while the linearity in $\boldsymbol{x}$ enables tight operator control and, ultimately, convergence guarantees for the reconstruction map.

A natural question is what $\mathcal{N}_\theta$ learns in (12). As illustrated in Figure 1, the network learns to assign high weights to regions that are similar under the chosen variants, so that the denoiser aggregates information from correlated structures. This kind of learned similarity reduces bias near discontinuities and improves denoising performance (Arias-Castro et al., 2012), as shown in Figure 1.

**Reconstruction Operator.** Our primary goal is to use the proposed denoiser for image reconstruction within a PnP framework. Among the available PnP schemes, we focus on PnP-HQS in (4). Plugging our denoiser into PnP-HQS yields the iteration

$$\boldsymbol{x}_{k+1} = \mathcal{D}\big(\mathrm{prox}_{\rho f}(\boldsymbol{x}_k); \boldsymbol{\xi}\big) \qquad (\rho > 0), \qquad (13)$$

where $\boldsymbol{\xi}$ is a reference image used to compute the aggregation weights. Equivalently, (13) is a fixed-point iteration $\boldsymbol{x}_{k+1} = \mathcal{T}_{\boldsymbol{\xi}}(\boldsymbol{x}_k)$ with reconstruction operator

$$\mathcal{T}_{\boldsymbol{\xi}} = \mathcal{D}(\,\cdot\,; \boldsymbol{\xi}) \circ \mathrm{prox}_{\rho f}. \qquad (14)$$

An important practical consideration is to find a reference image $\boldsymbol{\xi}$. This is in particular very important as $\boldsymbol{\xi}$ is passed through the trainable part, and if $\boldsymbol{\xi}$ is poor (e.g., at initialization), the learned weights can be unstable in early iterations. To address this, we use a short warm-up phase (see Algorithm 2). During warm-up, after each denoising step, we set the reference to the current iterate, which quickly yields a cleaner and more structured reference and improves the quality of the weight maps produced by $\mathcal{N}_\theta$. After the warm-up, we freeze the reference image for the remainder of the reconstruction. The effect of different warm-up lengths is described in Figure 5.

## 3. Theoretical Analysis

The fixed-point iteration induced by the denoiser in (12) does not, by itself, come with a convergence guarantee. To

obtain a contractive reconstruction map, we impose a small set of practical conditions on the permutation set $\mathcal{G}$, the network $\mathcal{N}_\theta$, and the forward operator $A$ in (1). Throughout, fix a reference image $\boldsymbol{\xi} \in \mathcal{X}$. Let $\mathcal{G}$ be a nonempty set of permutations of the lattice $\Omega$, and let $\mathcal{N}_\theta : \mathcal{X} \times \mathcal{X} \to \mathcal{X}$ be a parametric function. Let $\boldsymbol{e} \in \mathcal{X}$ denote the all-ones image, and let $\pi_{\mathrm{id}} \in \mathcal{G}$ denote the identity permutation, i.e., $\pi_{\mathrm{id}}(i) = i$ for all $i \in \Omega$. We make the following assumptions for our theoretical analysis:

(A1) *Set structure*: $\pi_{\mathrm{id}} \in \mathcal{G}$, and $\mathcal{G}$ is closed under inversion, i.e., if $\pi \in \mathcal{G}$, then $\pi^{-1} \in \mathcal{G}$.

(A2) *Positivity*: $\mathcal{N}_\theta(\boldsymbol{\xi}, \boldsymbol{\xi}') > 0$ for all $\boldsymbol{\xi}, \boldsymbol{\xi}' \in \mathcal{X}$.

(A3) *Inverse-Consistency Condition (ICC)*:

$$\forall \pi \in \mathcal{G} : \mathcal{N}_\theta\big(\boldsymbol{\xi}, \pi \cdot \boldsymbol{\xi}\big) = \pi \cdot \mathcal{N}_\theta\big(\boldsymbol{\xi}, \pi^{-1} \cdot \boldsymbol{\xi}\big). \quad (15)$$

(A4) *Transitivity*: For all $i, j \in \Omega$, there exist $\pi_1, \dots, \pi_k \in \mathcal{G}$ such that $\pi_k \cdots \pi_1(i) = j$.

(A5) *Nonannihilating forward model*: $A\boldsymbol{e} \neq 0$.

We can now state the main result.

**Theorem 3.1** (Contractivity of the reconstruction operator). *Let $\mathcal{D}$ be the denoiser defined in* (12). *Under assumptions* (A1)–(A5), *there exists a mapping $\varphi$ such that, for any $\boldsymbol{\xi} \in \mathcal{X}$, the fixed-point operator $\mathcal{T}_{\boldsymbol{\xi}}$ in* (13) *is contractive when the denoiser $\mathcal{D}$ is replaced by $\varphi(\mathcal{D})$.*

In the rest of this section, we analyze Assumptions (A1)–(A5). We begin with an immediate consequence of the positivity assumption on $\mathcal{N}_\theta$. Specifically, Assumption (A2) implies that the denoiser $\mathcal{D}(\cdot; \boldsymbol{\xi})$ defined in (12) has spectral radius equal to one.

**Proposition 3.2.** *Suppose $\mathcal{N}_\theta$ satisfies* (A2). *Then, for any $\boldsymbol{\xi} \in \mathcal{X}$, the denoiser $\mathcal{D}(\cdot; \boldsymbol{\xi})$ has spectral radius equal to* 1.

Although this spectral-radius constraint is important, it does not by itself imply that $\mathcal{D}(\cdot; \boldsymbol{\xi})$ is nonexpansive in the spectral norm $\|\cdot\|_2$. To obtain nonexpansivity, we use the remaining assumptions.

**Algorithm 1** Denoising Mechanism using (12) and ICC

**Input:** input image $\boldsymbol{x}$, reference image $\boldsymbol{\xi}$, permutations $\mathcal{G}$ (A1), Network $\mathcal{N}_\theta(\cdot, \cdot)$ (A2)
**Output:** $\mathcal{D}(\boldsymbol{x}; \boldsymbol{\xi})$
Partition $\mathcal{G}$ into disjoint sets $\mathcal{A}, \mathcal{B}, \mathcal{S}$ and $\{\pi_{\mathrm{id}}\}$ such that: $\mathcal{S}$ contains all order-2 elements and $\mathcal{A}$ and $\mathcal{B}$ are inverse pairs ($\pi^{-1} \in \mathcal{B}$ iff $\pi \in \mathcal{A}$)
Initialize $\mathcal{K} \leftarrow 0, C \leftarrow 0$
**for** each $\pi \in \mathcal{A}$ **do**
    Compute $w \leftarrow \mathcal{N}_\theta(\boldsymbol{\xi}, \pi \cdot \boldsymbol{\xi})$
    Compute inverse weight $w_{\mathrm{inv}} \leftarrow \pi^{-1} \cdot w$ (ICC)
    $\mathcal{K} \leftarrow \mathcal{K} + w \odot (\pi \cdot \boldsymbol{x})$
    $\mathcal{K} \leftarrow \mathcal{K} + w_{\mathrm{inv}} \odot (\pi^{-1} \cdot \boldsymbol{x})$
    $C \leftarrow C + w$
    $C \leftarrow C + w_{\mathrm{inv}}$
**end for**
Compute $w \leftarrow \mathcal{N}_\theta(\boldsymbol{\xi}, \boldsymbol{\xi})$
$\mathcal{K} \leftarrow \mathcal{K} + w \odot \boldsymbol{x}$
$C \leftarrow C + w$
**for** each $\pi \in \mathcal{S}$ **do**
    Continue
**end for**
**Return** $\mathcal{K}(\boldsymbol{x}; \boldsymbol{\xi}) \oslash C(\boldsymbol{\xi})$

---

**Algorithm 2** Reconstruction with $\varphi(\mathcal{D}) = \mathcal{D}_{\mathrm{sym}}$ and HQS

**Input:** initial image $\boldsymbol{x}_0$, step $\rho > 0$, warm-up iters $N_{\mathrm{warm}}$, reconstruction iters $N$
Set $\boldsymbol{\xi}_0 \leftarrow \boldsymbol{x}_0$.
**for** $k = 1$ to $N_{\mathrm{warm}}$ **do**
    $\boldsymbol{y}_k \leftarrow \mathrm{prox}_{\rho f}(\boldsymbol{x}_{k-1})$.
    $\boldsymbol{x}_k \leftarrow \mathcal{D}_{\mathrm{sym}}(\boldsymbol{y}_k; \boldsymbol{\xi}_{k-1})$.
    $\boldsymbol{\xi}_k \leftarrow \boldsymbol{x}_k$.
**end for**
Freeze reference $\boldsymbol{\xi} \leftarrow \boldsymbol{\xi}_{N_{\mathrm{warm}}}$.
**for** $k = N_{\mathrm{warm}} + 1$ to $N_{\mathrm{warm}} + N$ **do**
    $\boldsymbol{y}_k \leftarrow \mathrm{prox}_{\rho f}(\boldsymbol{x}_{k-1})$.
    $\boldsymbol{x}_k \leftarrow \mathcal{D}_{\mathrm{sym}}(\boldsymbol{y}_k; \boldsymbol{\xi})$.
**end for**
**Output:** reconstruction $\boldsymbol{x}_{N_{\mathrm{warm}}+N}$.

---

We first examine the implication of the inverse-consistency condition in Assumption (A3). We show that, when ICC holds, the operator $\mathcal{K}(\cdot\,; \boldsymbol{\xi})$ becomes symmetric for any fixed $\boldsymbol{\xi} \in \mathcal{X}$. Symmetry of this operator will play a central role in establishing the nonexpansiveness of the proposed denoiser and in establishing Theorem 3.1.

**Lemma 3.3.** *Fix $\boldsymbol{\xi} \in \mathcal{X}$. Suppose $\mathcal{G}$ satisfies (A1). The linear operator $\mathcal{K}(\cdot\,; \boldsymbol{\xi})$ is symmetric if $\mathcal{N}_\theta$ satisfies the ICC.*

The proof is purely combinatorial once we fix $\boldsymbol{\xi}$. We first identify the linear operator $\mathcal{K}(\cdot; \boldsymbol{\xi})$ with its matrix representation and reduce symmetry to showing that the $(i, j)$ and $(j, i)$ entries agree. Each entry is the sum over permutations that map one pixel index to the other. The condition (15) lets us rewrite the summand for a permutation $\pi$ in terms of $\pi^{-1}$, and the pullback action (7) then swaps the evaluation from index $i$ to index $j$. Since inversion is a bijection between the two relevant permutation sets, the two sums coincide. The full details are given in the appendix.

The practical implication of ICC (15) is that, for any black-box $\mathcal{N}_\theta$ and any reference image $\boldsymbol{\xi} \in \mathcal{X}$, the associated operator $\mathcal{K}(\cdot; \boldsymbol{\xi})$ can be made symmetric. *Importantly, this can be achieved without requiring any modification to the architecture or parameters of $\mathcal{N}_\theta$.* We defer the discussion of how ICC is enforced in practice to subsequent sections.

At this stage, ICC and positivity are the only structural conditions imposed on $\mathcal{N}_\theta$. Even under ICC and positivity, the normalized denoiser $\mathcal{D}(\cdot; \boldsymbol{\xi})$ need not be nonexpansive

in the spectral norm. The primary reason is that, although $\mathcal{K}(\cdot; \boldsymbol{\xi})$ can be made symmetric via ICC, the normalization used to form $\mathcal{D}(\cdot; \boldsymbol{\xi})$ generally destroys symmetry, in which case the spectral norm $\|\mathcal{D}(\cdot; \boldsymbol{\xi})\|_2$ can exceed 1.

However, we can apply a symmetrization protocol (Sreehari et al., 2016) to obtain a symmetric denoiser corresponding to $\mathcal{D}(\cdot;\, \boldsymbol{\xi})$. Define

$$\mathcal{D}_{\mathrm{sym}} = \varphi(\mathcal{D}), \tag{16}$$

where the function $\varphi$ transforms nonsymmetric $\mathcal{D}(\cdot;\, \boldsymbol{\xi})$ to a symmetric one:

$$\varphi(\mathcal{D})(\boldsymbol{x}; \boldsymbol{\xi}) = \frac{1}{\|\hat{\boldsymbol{e}}\|_\infty} C^{\frac{1}{2}} \mathcal{D} C^{-\frac{1}{2}}(\boldsymbol{x}) + \left( \boldsymbol{e} - \frac{\hat{\boldsymbol{e}}}{\|\hat{\boldsymbol{e}}\|_\infty} \right) \odot \boldsymbol{x}, \tag{17}$$

where, $C \equiv C(\boldsymbol{\xi})$ in (12) and we write $C^{\frac{1}{2}} \mathcal{D} C^{-\frac{1}{2}}(\boldsymbol{x}) \equiv C^{\frac{1}{2}} \odot \mathcal{D}(\boldsymbol{x} \oslash C^{\frac{1}{2}}; \boldsymbol{\xi})$ and $\hat{\boldsymbol{e}} \equiv C^{\frac{1}{2}} \mathcal{D} C^{-\frac{1}{2}}(\boldsymbol{e})$ for brevity. For any $\boldsymbol{\xi} \in \mathcal{X}$, $\varphi(\mathcal{D})(\cdot; \boldsymbol{\xi})$ is linear and under assumptions (A1)–(A3), it is symmetric, entrywise nonnegative, and its rows sum to one. These properties immediately imply the required nonexpansivity we are seeking.

**Lemma 3.4** (Nonexpansivity of the Denoiser). *Suppose (A1), (A2) and (A3) hold. Then $\mathcal{D}_{\mathrm{sym}}(\cdot; \boldsymbol{\xi})$ is symmetric, element-wise nonnegative, and stochastic. Consequently,*

$$\|\mathcal{D}_{\mathrm{sym}}(\cdot; \boldsymbol{\xi})\|_2 = 1. \tag{18}$$

Thus, the first three assumptions yield a nonexpansive denoiser while keeping $\mathcal{N}_\theta$ largely unconstrained. Assumption (A4) is imposed on the permutation set $\mathcal{G}$, whereas Assumption (A5) is imposed on the forward operator $A$ in (1). We use these last two assumptions to establish the contractivity of the reconstruction operator $\mathcal{T}_{\boldsymbol{\xi}}$ and conclude the proof of Theorem 3.1, with $\varphi$ being the mapping defined in (17). The details are provided in the Appendix.

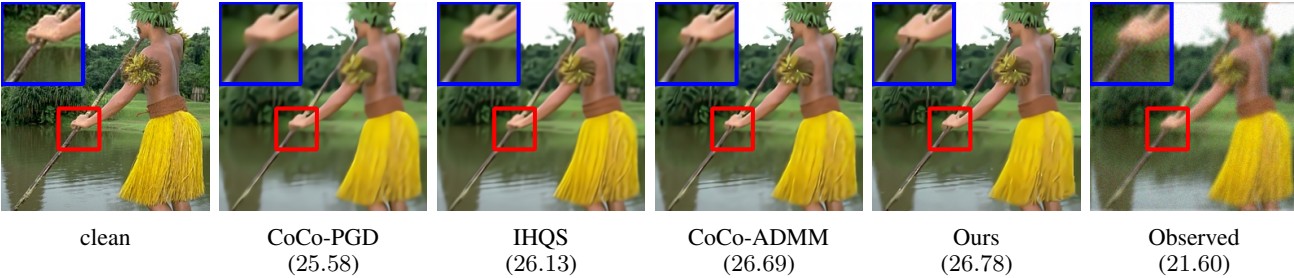

| clean | CoCo-PGD (25.58) | IHQS (26.13) | CoCo-ADMM (26.69) | Ours (26.78) | Observed (21.60) |

*Figure 3.* Visual comparison on the "man" image under motion blur (kernel 3) and noise level $\nu = 0.03$. Our denoiser suppresses artifacts while avoiding oversmoothing more effectively than CoCo-DRUNet (Wei et al., 2025) and IHQS (Wei et al., 2024).

## 4. Implementation

Algorithm 1 implements the denoiser in (12) by aggregating permuted versions of the input image. The construction is designed to satisfy Assumptions (A1)–(A3). Given an input image $x$ and a reference image $\xi$, we apply the chosen permutations to form variants $\pi \cdot x$ and $\pi \cdot \xi$. For each $\pi$, we run a lightweight CNN $\mathcal{N}_\theta$ on $(\xi, \pi \cdot \xi)$ to produce a positive weight map $w$. A key implementation detail is that we do not process $\pi$ and $\pi^{-1}$ independently. Instead, we impose ICC directly inside Algorithm 1 by generating the weight for $\pi^{-1}$ from the weight for $\pi$: $w_{\mathrm{inv}} = \pi^{-1} \cdot w$, which is a cheap operation and avoids an extra forward pass through $\mathcal{N}_\theta$. In practice, this reduces computation and ensures that the resulting denoiser satisfies the structural conditions used in our theoretical analysis.

The implementation maintains two buffers: a numerator that sums weighted images, and a denominator that sums the weights. After looping over all permutations, we divide the two element-wise to obtain the final output. A key design choice in our construction is the set of permutations used to generate image variants. Natural candidates include translations on the image lattice, the dihedral group $D_4$ (rigid symmetries of the square), and diagonal translations.

**Order-$2$ elements.** The ICC condition interacts in a special way with order-2 elements. If $\mathcal{G}$ contains a nontrivial permutation $\pi$ with $\pi = \pi^{-1}$, then ICC reduces to

$$\mathcal{N}_\theta\bigl(\xi, \pi \cdot \xi\bigr) = \pi \cdot \mathcal{N}_\theta\bigl(\xi, \pi \cdot \xi\bigr). \tag{19}$$

This requires the output $\mathcal{N}_\theta(\xi, \pi \cdot \xi)$ to be *invariant* under $\pi$ for every $\xi$. For a generic image $\xi$ and a nontrivial $\pi$, such invariance is overly restrictive and is not something we can reasonably enforce during training. Therefore, relying on ICC alone is not suitable when $\mathcal{G}$ contains many order-2 elements. For example, the dihedral group $D_4$ includes several elements (e.g., a $180°$ rotation and reflections), and hence is not a good fit for our ICC-based symmetry construction.

In practice, given a set of permutations $\mathcal{G}$ which is closed

under inversion, we can identify the subset of involutions

$$\mathcal{S} = \{\pi \in \mathcal{G} : \pi = \pi^{-1}, \ \pi \neq \pi_{\mathrm{id}}\}, \tag{20}$$

and exclude these elements when enforcing ICC (see Algorithm 1).

A convenient alternative is to use translations, which avoid the involution issue but are typically too large to use in full. For efficiency, we therefore restrict to a *local* set of translations within a radius $R$, denoted by $T[R]$. This choice is closed under inversion and contains no nontrivial order-2 elements. Moreover, translations act transitively on the lattice. As a result, $T[R]$ satisfies the permutation assumptions in Section 3. On the other hand, the forward operators for all major linear inverse problems, including deblurring and superresolution, satisfy the *Nonannihilating* condition, yielding a contractive reconstruction operator.

**Training** We can use any network architecture for $\mathcal{N}_\theta$ in Algorithm 1, followed by a positive activation. For the reconstruction experiments in Tables 1 and 2, we use a lightweight CNN for $\mathcal{N}_\theta$ (with 0.9M parameters) and optionally concatenate a constant noise-level channel $\sigma$ to the input. When included, this channel makes the denoiser noise-aware and serves as a hyperparameter during reconstruction. The network is trained implicitly by minimizing the MSE loss for the denoiser $\mathcal{D}$ in Algorithm 1. Training samples are generated on the fly from clean images $x_g$ by adding Gaussian noise with $\sigma \sim \mathcal{U}(0, 50/255)$. To better match reconstruction-time behavior, we use a noisier reference image by adding extra noise with standard deviation $\sigma_z = \delta\sigma$, with $\delta = 1.2$.

## 5. Experiments

We evaluate the performance of our proposed denoiser in terms of denoising (Algorithm 1) and as an implicit regularizer in reconstructions (Algorithm 2). We use $T[7]$ as the underlying set of permutations in all experiments. In Figure 2, we compare our method with BM3D (Dabov et al., 2007) and the unconstrained DRUNet (Zhang et al., 2021)

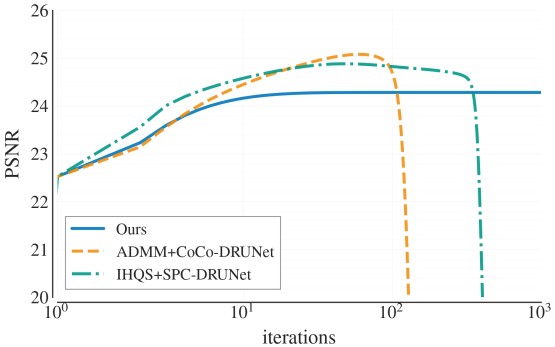

*Figure 4.* Instability of IHQS+SPC-DRUNet (Wei et al., 2024) and CoCo-ADMM (Wei et al., 2025) in $3\times$ superresolution PnP framework of *coral* image of CBSD68. The corresponding norm plot is provided in the Appendix.

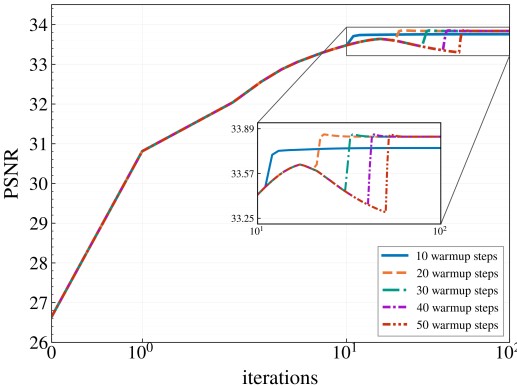

*Figure 5.* Adaptive warmup ablation of Algorithm 2. We demonstrate the convergence of our proposed method with different warmups, then freeze the reference. The inset zooms in on the final iterations to highlight the effect of different warm-up lengths.

at a high noise level of $50/255$. Our denoiser outperforms BM3D by a large margin and remains close to DRUNet despite being provably nonexpansive. Qualitatively, it preserves fine details well, which we attribute to the learned $\mathcal{N}_\theta$ that produces sharper, more selective aggregation weights. Additional results across datasets and noise levels are reported in Appendix Table 4 and show the same trend.

We next evaluate our Contractive PnP performance on standard deblurring and superresolution tasks with the symmetrized denoiser $\mathcal{D}_{\text{sym}}$ using Algorithm 2. The output of $\mathcal{D}_{\text{sym}}$ is obtained by transforming Algorithm 1 through $\varphi$ in (17). Throughout, when referring to a baseline PnP framework, we use the notation Algo+Denoiser, where Algo is the underlying iterative algorithm. In all experiments, we compare with recent *convergent* PnP algorithms, namely, ADMM+CoCo-DRUNet (Wei et al., 2025), PGD+CoCo-DRUNet (Wei et al., 2025), IHQS+SPC-DRUNet (Wei et al., 2024) (Algorithm 2), GSPnP+GSDRUNet (Hurault et al., 2022a), SAGD+WCRR (Goujon et al., 2024),

HQS+LipDSNN (Ducotterd et al., 2024) along with the classical options FBS+TV (Rudin et al., 1992) and ADMM+DSG-NLM (Sreehari et al., 2016). For all competing algorithms, we use the default parameter settings recommended in their respective papers, ensuring faithful reproduction of the originally reported behavior.

Tables 1 and 2 report PSNR on CBSD10 for blur kernels, scale factors, and noise levels. In both deblurring and superresolution, our method is competitive with the convergent baselines for both $\nu = 0.03$ and $\nu = 0.05$. A visual comparison is shown in Figure 3.

A key observation is that the Ishikawa-style method IHQS+SPC-DRUNet (Wei et al., 2024) and the CoCo-DRUNet based ADMM/PGD variants (Wei et al., 2025) may show *divergence* in some cases, despite their stated convergence guarantees (Figure 4). This behavior stems from the fact that both SPC and CoCo variants of DRUNet rely on *spectral regularization of the denoiser Jacobian solely on the training set*, which enforces pseudo-contractivity only empirically. Such sample-limited constraints do not guarantee the desired constraints outside the training distribution, allowing iterates to drift into uncontrolled regions and diverge. In contrast, because our method imposes constraints globally, our denoiser remains stable across all tested scenarios without requiring these penalties, offering performance on par with or better than such methods while providing substantially stronger theoretical guarantees.

Figure 5 demonstrates the role of the warm-up length in the reconstruction procedure in Algorithm 2. The convergence analysis in this work is established for a fixed operator $\mathcal{T}_{\boldsymbol{\xi}}$. If $\boldsymbol{\xi}$ were updated throughout reconstruction, the operator itself would change across iterations, and the contraction guarantee in Theorem 3.1 would no longer apply. Since the reference controls the weights through the nonlinear network $\mathcal{N}_\theta$, a poor reference, such as the initialization, can lead to poor learned weights. Thus, the warm-up stage in Algorithm 2 provides a practical way to improve the reference while preserving the theoretical setting needed for convergence. As shown in Figure 5, *the method is not very sensitive to the exact warm-up length*. A warm-up of around 20 iterations is a good default, and the performance remains stable over a fairly broad range. At the same time, the warm-up should not be made too long, since excessively delaying the frozen-reference phase can reduce the final reconstruction quality; the contraction guarantee applies only after the reference is fixed.

**Computation Time.** The computation time of our method scales with the number of permutations because of the aggregation step. Thus, using a smaller $\mathcal{G}$ leads to faster reconstruction. The set $T[7]$ contains 225 permutations, over which we evaluate the neural network $\mathcal{N}_\theta$ in (12). The ICC strategy in Algorithm 1 improves efficiency by roughly halv-

*Table 1.* PSNR (dB) comparison of image deblurring methods on CBSD10 across nine blur kernels and noise levels $\nu \in \{0.03, 0.05\}$. Best scores are in **bold**, second best are underlined, and our method is highlighted in orange.

| $\nu$ | Method | | | | | | | | | | *Avg* |
|---|---|---|---|---|---|---|---|---|---|---|---|
| 0.03 | Observed | 21.48 | 20.96 | 21.68 | 17.60 | 22.06 | 17.92 | 18.89 | 18.76 | 23.48 | 20.32 |
| | ADMM+CoCo-DRUNet (Wei et al., 2025) | 28.74 | 28.29 | 28.46 | 27.98 | 29.81 | 29.70 | 28.71 | 28.28 | 27.55 | 28.61 |
| | ADMM+DSG-NLM (Sreehari et al., 2016) | 24.91 | 24.73 | 25.53 | 24.30 | 26.43 | 25.71 | 25.48 | 24.88 | 25.92 | 25.32 |
| | FBS+CoCo-DRUNet (Wei et al., 2025) | 27.28 | 26.79 | 27.14 | 26.49 | 28.49 | 28.49 | 27.55 | 27.12 | 26.97 | 27.37 |
| | FBS+TV (Rudin et al., 1992) | 25.79 | 25.62 | 26.40 | 25.37 | 27.29 | 26.90 | 26.46 | 25.99 | 26.48 | 26.26 |
| | GSPnP+GSDRUNet (Hurault et al., 2022a) | **29.17** | **28.85** | **29.14** | **28.58** | **29.94** | **30.10** | **29.28** | **28.90** | **27.85** | **29.09** |
| | IHQS+SPC-DRUNet (Wei et al., 2024) | 27.85 | 27.28 | 27.61 | 26.88 | 29.12 | 29.04 | 27.90 | 27.31 | 26.95 | 27.77 |
| | HQS+LipDSNN (Ducotterd et al., 2024) | 26.05 | 25.98 | 26.63 | 25.68 | 27.43 | 27.01 | 26.59 | 26.14 | 26.41 | 26.43 |
| | SAGD+WCRR (Goujon et al., 2024) | 26.82 | 26.50 | 27.13 | 26.35 | 28.29 | 27.96 | 27.36 | 26.93 | 26.95 | 27.14 |
| | Ours | 28.23 | 27.89 | 28.32 | 27.57 | 29.48 | 29.25 | 28.46 | 28.01 | 27.44 | 28.29 |
| 0.05 | Observed | 20.45 | 20.03 | 20.60 | 17.14 | 20.90 | 17.43 | 18.28 | 18.17 | 22.00 | 19.44 |
| | ADMM+CoCo-DRUNet (Wei et al., 2025) | 27.12 | 26.95 | 27.28 | 26.61 | 28.27 | 28.05 | **27.39** | **26.99** | **26.91** | 27.29 |
| | ADMM+DSG-NLM (Sreehari et al., 2016) | 23.63 | 23.57 | 24.43 | 23.27 | 24.95 | 24.37 | 24.14 | 23.62 | 24.82 | 24.09 |
| | FBS+CoCo-DRUNet (Wei et al., 2025) | 26.15 | 25.72 | 26.39 | 25.46 | 27.58 | 27.24 | 26.67 | 26.15 | 26.57 | 26.44 |
| | FBS+TV (Rudin et al., 1992) | 24.45 | 24.26 | 25.20 | 23.96 | 25.97 | 25.38 | 25.03 | 24.45 | 25.56 | 24.92 |
| | GSPnP+GSDRUNet (Hurault et al., 2022a) | **27.36** | **27.16** | **27.61** | **26.91** | **28.55** | **28.27** | 27.32 | **26.99** | 26.71 | **27.43** |
| | IHQS+SPC-DRUNet (Wei et al., 2024) | 26.73 | 26.20 | 26.67 | 25.91 | 27.90 | 27.76 | 26.92 | 26.44 | 26.54 | 26.79 |
| | HQS+LipDSNN (Ducotterd et al., 2024) | 24.45 | 24.40 | 25.22 | 24.09 | 25.78 | 25.20 | 25.04 | 24.61 | 25.61 | 24.93 |
| | SAGD+WCRR (Goujon et al., 2024) | 25.45 | 25.20 | 25.97 | 24.89 | 26.84 | 26.45 | 25.94 | 25.54 | 26.12 | 25.82 |
| | Ours | 26.42 | 26.28 | 26.87 | 25.95 | 27.82 | 27.47 | 26.88 | 26.49 | 26.62 | 26.75 |

*Table 2.* PSNR (dB) comparison of superresolution methods on CBSD10 for scale factors $s \in \{2, 3, 4\}$ and noise levels $\nu \in \{0.03, 0.05\}$. Best entries are **bold**, second best are underlined, and our method is highlighted in orange.

| Kernels | Method | $s = 2$ | | $s = 3$ | | $s = 4$ | | *Avg* |
|---|---|---|---|---|---|---|---|---|
| | | $\nu = 0.03$ | $\nu = 0.05$ | $\nu = 0.03$ | $\nu = 0.05$ | $\nu = 0.03$ | $\nu = 0.05$ | |
| | Bicubic | 23.74 | 22.49 | 22.28 | 21.32 | 21.10 | 20.33 | 21.88 |
| | ADMM+CoCo-DRUNet (Wei et al., 2025) | 26.94 | **26.19** | 25.37 | **24.87** | 24.08 | 23.72 | 25.20 |
| | ADMM+DSG-NLM (Sreehari et al., 2016) | 25.00 | 23.94 | 23.99 | 23.02 | 23.18 | 22.31 | 23.57 |
| | FBS+CoCo-DRUNet (Wei et al., 2025) | 26.75 | 25.26 | 24.15 | 21.64 | 22.52 | 19.64 | 23.33 |
| | FBS+TV (Rudin et al., 1992) | 25.59 | 24.37 | 24.53 | 23.60 | 23.51 | 22.81 | 24.07 |
| | GSPnP+GSDRUNet (Hurault et al., 2022a) | **27.09** | 26.12 | **25.82** | 24.66 | **24.50** | **23.93** | **25.35** |
| | IHQS+SPC-DRUNet (Wei et al., 2024) | 26.61 | 25.98 | 25.22 | 24.73 | 23.91 | 23.60 | 25.01 |
| | HQS+LipDSNN (Ducotterd et al., 2024) | 25.11 | 24.71 | 24.19 | 23.57 | 23.27 | 22.63 | 23.91 |
| | SAGD+WCRR (Goujon et al., 2024) | 26.18 | 25.19 | 24.73 | 23.87 | 23.54 | 22.79 | 24.38 |
| | Ours | 26.77 | 25.86 | 25.21 | 24.49 | 23.96 | 23.41 | 24.95 |

ing the aggregation loop, from 225 to 113 evaluations. The

*Table 3.* Computation time per iteration for $\mathcal{G} = T[7]$.

| Method | Algorithm 1 | $\mathcal{D}_{\text{sym}}$ (direct) | $\mathcal{D}_{\text{sym}}$ (cached) |
|---|---|---|---|
| Time/iter. (s) | 0.46 | 1.39 | 0.94 |

symmetrization step in Equation (17) further increases the runtime, since it requires running Algorithm 1 and evaluating $\mathcal{N}_\theta$ three times for a single denoising step. In practice, this overhead can be reduced by caching the outputs of $\mathcal{N}_\theta$ and reusing them, avoiding repeated inference through $\mathcal{N}_\theta$. For $\mathcal{G} = T[7]$, the computation times are shown in Table 3. The code for the implementation is available at github.com/arghyasinha/nectr.

## 6. Conclusion

We proposed a new way to construct a provably nonexpansive denoiser, parameterized by a lightweight CNN. Under practical, mild assumptions, we established rigorous guarantees of nonexpansivity. Moreover, when plugged into PnP-HQS, our denoiser yields a *contractive* reconstruction operator, which is a particularly strong form of convergence guarantee. Despite these global constraints, the resulting reconstruction system remains expressive and achieves performance competitive with recent convergent baselines. Finally, unlike approaches that enforce Lipschitz control only empirically and may diverge as shown in Section 5, our guarantees hold globally and rule out divergence, with convergence to a unique fixed point due to contraction.

## Impact Statement

This work aims to advance reliable machine learning methods for computational imaging. The proposed framework is interpretable by design and provides convergence guarantees for reconstruction algorithms based on trainable denoisers, thereby reducing the risk of divergence in inverse imaging problems. These guarantees support the development of robust and trustworthy reconstruction systems in applications where reliability is essential.

## Acknowledgements

A. Sinha was supported by the Government of India through the Prime Minister Research Fellowship (PMRF) TF/PMRF-22-5534 and by Qualcomm Technologies, Inc. through the Qualcomm Innovation Fellowship India 4300074105. K. N. Chaudhury was supported by the Government of India through grant ANRF/ARG/2025/00696/ENS. The authors thank the Kotak IISc AI-ML Centre for providing GPU resources.

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

# Appendix

**Proof of Proposition 3.2.**

Assumption A2 implies that $\mathcal{D}(\cdot; \boldsymbol{\xi})$ is represented by an entry-wise nonnegative matrix. Moreover, since $\mathcal{D}(\cdot; \boldsymbol{\xi})$ is obtained by normalizing $\mathcal{K}(\cdot; \boldsymbol{\xi})$ element-wise, its rows sum to one, as shown below.

Substituting $\boldsymbol{x} = \boldsymbol{e}$ into (11) gives

$$\mathcal{K}(\boldsymbol{e}; \boldsymbol{\xi}) = \sum_{\pi \in \mathcal{G}} \mathcal{N}_\theta(\boldsymbol{\xi}, \pi \cdot \boldsymbol{\xi}) \odot (\pi \cdot \boldsymbol{e}). \tag{21}$$

Since permutations leave the all-ones image invariant, we have $\pi \cdot \boldsymbol{e} = \boldsymbol{e}$ for all $\pi \in \mathcal{G}$. Therefore,

$$\mathcal{K}(\boldsymbol{e}; \boldsymbol{\xi}) = \sum_{\pi \in \mathcal{G}} \mathcal{N}_\theta(\boldsymbol{\xi}, \pi \cdot \boldsymbol{\xi}) = C, \tag{22}$$

where $C$ is the normalization factor in the definition of $\mathcal{D}(\cdot; \boldsymbol{\xi})$. Applying the normalization yields

$$\mathcal{D}(\boldsymbol{e}; \boldsymbol{\xi}) = \boldsymbol{e}. \tag{23}$$

Thus, $1$ is an eigenvalue of $\mathcal{D}(\cdot; \boldsymbol{\xi})$, and hence the spectral radius $\rho$ satisfies the following:

$$\rho\big(\mathcal{D}(\cdot; \boldsymbol{\xi})\big) \geqslant 1. \tag{24}$$

On the other hand, since $\mathcal{D}(\cdot; \boldsymbol{\xi})$ is entry-wise nonnegative and satisfies $\mathcal{D}(\boldsymbol{e}; \boldsymbol{\xi}) = \boldsymbol{e}$, we have,

$$\|\mathcal{D}(\cdot; \boldsymbol{\xi})\|_\infty = 1. \tag{25}$$

Since the spectral radius is bounded above by any induced matrix norm, we obtain

$$\rho\big(\mathcal{D}(\cdot; \boldsymbol{\xi})\big) \leqslant \|\mathcal{D}(\cdot; \boldsymbol{\xi})\|_\infty = 1. \tag{26}$$

Combining the two inequalities gives

$$\rho\big(\mathcal{D}(\cdot; \boldsymbol{\xi})\big) = 1. \tag{27}$$

This proves the claim. $\qquad\square$

**Proof of Lemma 3.3.**

Fix $\boldsymbol{\xi} \in \mathcal{X}$. Since $\mathcal{K}(\cdot\,; \boldsymbol{\xi})$ is linear in its first argument, it can be identified with a matrix indexed by $\Omega \times \Omega$. To show symmetry, it suffices to verify that

$$\mathcal{K}(\cdot\,; \boldsymbol{\xi})_{ij} = \mathcal{K}(\cdot\,; \boldsymbol{\xi})_{ji} \tag{28}$$

for all $i, j \in \Omega$. For fixed $i, j \in \Omega$, define

$$\mathcal{P}_{ij} := \{\pi \in \mathcal{G} \mid \pi(i) = j\}.$$

Expanding the definition of $\mathcal{K}(\cdot\,; \boldsymbol{\xi})$ using (11) gives

$$\mathcal{K}(\cdot\,; \boldsymbol{\xi})_{ij} = \sum_{\pi \in \mathcal{P}_{ji}} \mathcal{N}_\theta(\boldsymbol{\xi}, \pi \cdot \boldsymbol{\xi})(i). \tag{29}$$

Applying the ICC (A3) transforms the expression of right side into

$$\sum_{\pi \in \mathcal{P}_{ji}} \big(\pi \cdot \mathcal{N}_\theta(\boldsymbol{\xi}, \pi^{-1} \cdot \boldsymbol{\xi})\big)(i).$$

Using the pullback action defined in (7), this can be rewritten as

$$\sum_{\pi \in \mathcal{P}_{ji}} \mathcal{N}_\theta(\boldsymbol{\xi}, \pi^{-1} \cdot \boldsymbol{\xi})\big(\pi^{-1}(i)\big).$$

Since $\pi \in \mathcal{P}_{ji}$ implies $\pi^{-1}(i) = j$, the above reduces to

$$\sum_{\pi \in \mathcal{P}_{ji}} \mathcal{N}_\theta(\boldsymbol{\xi}, \pi^{-1} \cdot \boldsymbol{\xi})(j).$$

Finally, observing that the map $\pi \mapsto \pi^{-1}$ is a bijection from $\mathcal{P}_{ji}$ to $\mathcal{P}_{ij}$, the expression becomes

$$\sum_{\pi \in \mathcal{P}_{ij}} \mathcal{N}_\theta(\boldsymbol{\xi}, \pi \cdot \boldsymbol{\xi})(j),$$

which is exactly the $(j, i)$ entry. This establishes symmetry of $\mathcal{K}(\cdot\,;\boldsymbol{\xi})$. $\qquad\square$

**Proof of Lemma 3.4.**

Fix $\boldsymbol{\xi} \in \mathcal{X}$. $\mathcal{D}_{\mathrm{sym}}(\cdot\,;\boldsymbol{\xi})$ is the sum of two terms in (17). For the first term, note that Lemma 3.3 implies $\mathcal{K}(\cdot\,;\boldsymbol{\xi})$ is symmetric. Thus $C^{\frac{1}{2}} \odot \mathcal{D}(\boldsymbol{x} \oslash C^{\frac{1}{2}};\boldsymbol{\xi}) = C^{-\frac{1}{2}} \odot \mathcal{K}(\boldsymbol{x} \odot C^{-\frac{1}{2}};\boldsymbol{\xi})$ is also symmetric.

The second term is symmetric trivially. Hence $\mathcal{D}_{\mathrm{sym}}(\cdot\,;\boldsymbol{\xi})$ is symmetric.

(A2) implies the nonnegativity of the first term. Moreover, $\hat{e} = C^{\frac{1}{2}} \mathcal{D} C^{-\frac{1}{2}} e \geqslant 0$, so the term $(e - \|\hat{e}\|_\infty^{-1}\hat{e})$ is also element-wise nonnegative, implying $\mathcal{D}_{\mathrm{sym}}(\cdot\,;\boldsymbol{\xi}) \geqslant 0$.

Finally, stochasticity follows from evaluating on $e$: the first term maps $e$ to $\|\hat{e}\|_\infty^{-1}\hat{e}$ by definition of $\hat{e}$, while the diagonal term maps $e$ to $e - \|\hat{e}\|_\infty^{-1}\hat{e}$, so their sum equals $e$.

Since $\mathcal{D}_{\mathrm{sym}}(\cdot\,;\boldsymbol{\xi})$ is symmetric and satisfies $\mathcal{D}_{\mathrm{sym}}(e;\boldsymbol{\xi}) = e$, it has eigenvalue 1. Together with element-wise nonnegativity and stochasticity, all eigenvalues lie in $[-1, 1]$, and therefore $\|\mathcal{D}_{\mathrm{sym}}(\cdot\,;\boldsymbol{\xi})\|_2 = 1$. $\qquad\square$

**Proof of Theorem 3.1.**

Since

$$f(\boldsymbol{x}) = \frac{1}{2}\|A\boldsymbol{x} - \boldsymbol{y}\|^2,$$

The proximal map has the affine form

$$\mathrm{prox}_{\rho f}(\boldsymbol{x}) = (I + \rho A^\top A)^{-1}\boldsymbol{x} + \text{affine term}. \tag{30}$$

The affine term does not affect contractiveness. Hence, for the contraction analysis, it is enough to consider the linear part.

For brevity, let

$$\phi(\mathcal{D}) = \mathcal{D}_{\mathrm{sym}} \equiv \mathcal{D}_{\mathrm{sym}}(\cdot\,;\boldsymbol{\xi}), \qquad \mathbf{E} := (I + \rho A^\top A)^{-1}.$$

Thus, proving contractiveness of (13) with denoiser $\mathcal{D}_{\mathrm{sym}}$ reduces to showing

$$\|\mathcal{D}_{\mathrm{sym}}\mathbf{E}\|_2 < 1.$$

From Lemma 3.4, the eigenvalues of $\mathcal{D}_{\mathrm{sym}}$ are real and lie in $[-1, 1]$. We order them according to their squared magnitudes:

$$1 = \lambda_1^2 \geqslant \lambda_2^2 \geqslant \cdots \geqslant \lambda_{|\Omega|}^2 \geqslant 0. \tag{31}$$

Let $\boldsymbol{u}$ be a unit eigenvector corresponding to $\lambda_1 = 1$. Using the spectral decomposition (Horn & Johnson, 2012) of $\mathcal{D}_{\mathrm{sym}}$, any vector can be decomposed into components parallel and orthogonal to $\boldsymbol{u}$. This gives

$$\|\mathcal{D}_{\mathrm{sym}}\mathbf{E}\|_2^2 \leqslant (\lambda_1^2 - \lambda_2^2)\|\mathbf{E}\boldsymbol{u}\|_2^2 + \lambda_2^2. \tag{32}$$

Hence, it is enough to prove that

$$\lambda_2^2 < \lambda_1^2 = 1 \qquad \text{and} \qquad \|\mathbf{E}\boldsymbol{u}\|_2^2 < 1.$$

We first prove $\lambda_2^2 < 1$. Since $\mathcal{D}_{\text{sym}}$ is stochastic, $e$ is an eigenvector corresponding to the eigenvalue $\lambda_1 = 1$. Moreover, since $\mathcal{G}$ is transitive (A4), (29) implies that $\mathcal{D}_{\text{sym}}$ is irreducible. Therefore, by the Perron–Frobenius theorem (Horn & Johnson, 2012), the eigenvalue 1 is simple, and the corresponding unit eigenvector is

$$u = \frac{e}{\|e\|_2}.$$

Thus, no other eigenvalue of $\mathcal{D}_{\text{sym}}$ other than $\lambda_1$ can be equal to 1. Hence $\lambda_2 \neq 1$. In addition, since $\pi_{\text{id}} \in \mathcal{G}$ using (A1), (29) gives $\mathcal{K}_{ii} > 0$ for all $i$. Hence $\mathcal{D}_{\text{sym}}$ has a positive diagonal. Perron–Frobenius theory implies that $-1$ is not an eigenvalue. Thus $\lambda_2 \neq -1$. Consequently,

$$\lambda_2^2 < 1.$$

It remains to show that $\|\mathbf{E}u\|_2^2 < 1$. By construction, $\mathbf{E}$ is symmetric positive definite and all its eigenvalues lie in $(0, 1]$. Hence

$$\|\mathbf{E}u\|_2 \leqslant 1.$$

Suppose $\|\mathbf{E}u\|_2 = 1$. Since $\mathbf{E}$ is symmetric positive definite with eigenvalues at most 1, this can happen if and only if $\mathbf{E}u = u$ which is not possible due to (A5). Therefore,

$$\|\mathbf{E}u\|_2^2 < 1.$$

Combining $\lambda_2^2 < 1$ with $\|\mathbf{E}u\|_2^2 < 1$ in (32) yields

$$\|\mathcal{D}_{\text{sym}}\mathbf{E}\|_2^2 < 1.$$

Thus $\|\mathcal{D}_{\text{sym}}\mathbf{E}\|_2 < 1$. $\qquad\square$

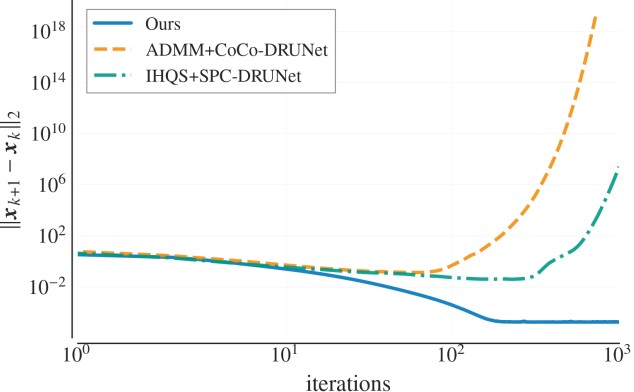

*Figure 6.* Instability IHQS+SPC-DRUNet (Wei et al., 2024) and CoCo-ADMM (Wei et al., 2025) in $3\times$ superresolution PnP framework of *coral* image of CBSD68.

*Table 4.* PSNR (dB) for CBSD68 across different noise sigmas

| Sigma | Noisy | Ours | BM3D | DRUNet |
|---|---|---|---|---|
| 10 | 28.1317 | 35.8099 | 33.3709 | 36.5875 |
| 15 | 24.6089 | 33.6400 | 31.1812 | 34.3515 |
| 20 | 22.1108 | 32.1441 | 29.7367 | 32.8396 |
| 25 | 20.1711 | 31.0265 | 28.6792 | 31.7277 |
| 50 | 14.1511 | 27.7191 | 25.7702 | 28.5335 |

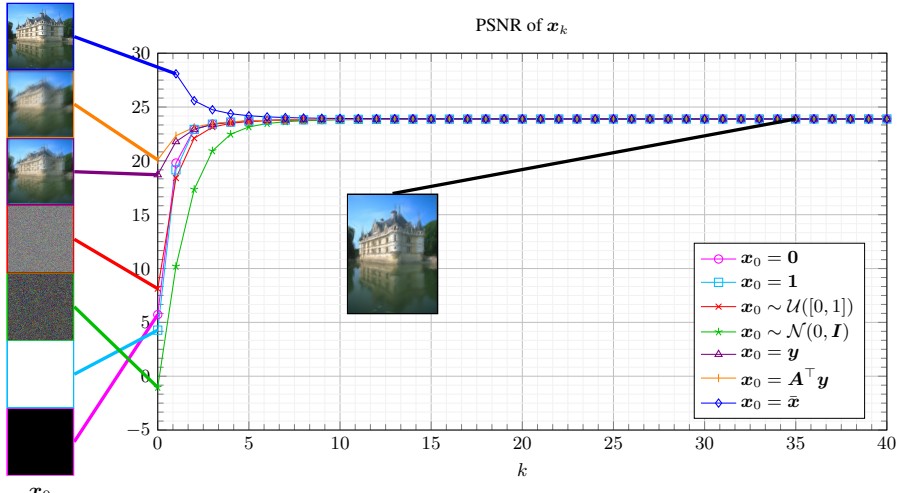

$\boldsymbol{x}_0$

*Figure 7.* **Consequence of Contractivity:** Motion deblurring task on the *castle* image under blur kernel 4 with noise level $\nu = 0.01$. For all initializations, the iterates converge to the same reconstruction. This illustrates the independence of Algorithm 2 from the initialization $\boldsymbol{x}_0$, for a fixed reference $\boldsymbol{\xi}$, owing to the contractivity of Algorithm 2.

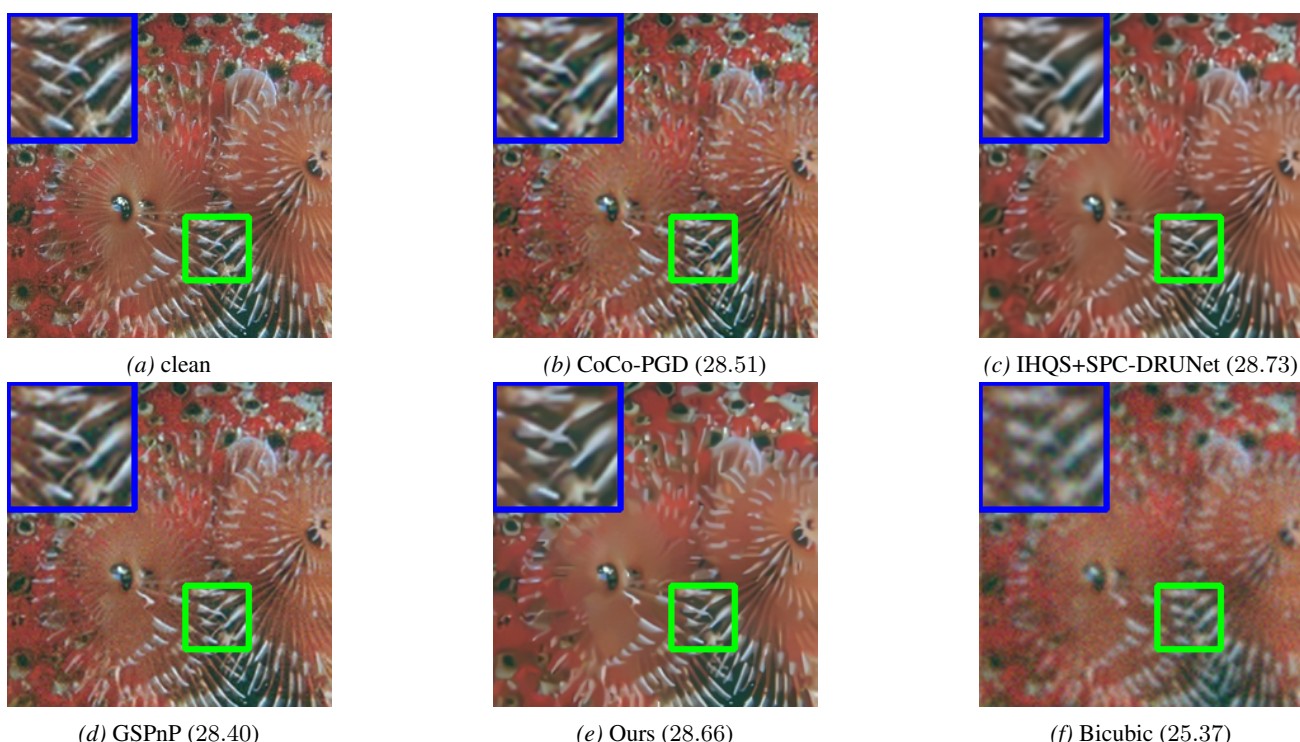

*Figure 8.* Superresolution ($2\times$) of the "coral" image (CBSD68) under kernel 1 and noise level $\nu = 0.01$.

