# OpenReview forum: "Trainable Nonexpansive Denoisers for Contractive Image Reconstruction"
_ICML.cc/2026/Conference — ICML 2026 regular_

### Official Review · Reviewer_6uJm · 2026-03-11

**Soundness:** 2
**Presentation:** 2
**Significance:** 3
**Originality:** 3
**Overall Recommendation:** 4
**Confidence:** 4

**Summary:**

This paper deals with the definition neural network architectures that are non expensive, in order to be used in a Plug-and-Play (PnP) framework.

**Compliance With Llm Reviewing Policy:**

Affirmed.

**Final Justification:**

The reviewer thinks that the paper is interesting and the authors gave interesting details in the rebuttal that enhance the paper quality.
This is why I increased my score.

**Key Questions For Authors:**

Some specific remarks:
1) Figure 1, some descriptions is missing on what we observe on the left/right plots.
2) after (9) what is e ?
3) In pratice, what is taken for the warm up procedure (algo 2) for x_0 ?
4) Figure 2, add names (a) (b) (c) (d) to the plots and add zooms to help the reader catching the differences.
5) In algorithm 1, the network N_\eta takes 3 parameters while it is defined only with 2 (see (7)).
This appears at different places. This is almost surely a problem of notations. But please correct.
And anyway, what is \sigma ?
6) Concerning the results reported in Tables 1 and 2.
Their analysis is made twice page 8 in the first anf second column, and this should be grouped.
The authors said either that "The proposed method consistently ranks among the top two performers" which is not true as the "ours" score is never bold nore underlined.
The second paragraph stating "Our method consistently ranks among the top methods" in more in line, yet not really informative.
7) The performances are only evaluated in terms of reconstruction quality (which is important of course) but what about the compelexity (number of parameters,...), computation time ?

Minor comments: there are a couple of typos, e.g., "Section Section 4" in Section 2 page 3.
The paper needs to be proofread.

**Limitations:**

yes

**Strengths And Weaknesses:**

On the positive aspects, this is a very interesting and timely topic. The objective of developing stable PnP is a very important topic.

However, on the negative aspects, the constrained PnP obtain lower performance, which may seem not really negligible.
The writting is not always good and the functions are not always well defined.

---

> ### Author Rebuttal · Authors · 2026-03-30
>
> We sincerely thank the reviewer for the careful reading of our paper and for recognizing the importance of developing stable PnP methods. We are also very grateful for the constructive comments on clarity and evaluation, which help us improve the presentation of our work. We will carefully incorporate these valuable suggestions in the final version. Below, we address each point in detail. We greatly appreciate your time and consideration, and we look forward to your final evaluation.
>
> ## (C1) Reply to the Weaknesses
>
> Our goal is to establish global nonexpansivity of the denoiser and provable contractivity of the reconstruction operator, which is a substantially stronger guarantee than those provided by the baselines. Despite this stronger requirement, our method remains competitive overall and outperforms the baselines in many cases (Fig. 3), while offering these added theoretical guarantees.
>
> ## Clarifications
>
> ### (C2) Fig. 1 (missing description)
>
> Each panel shows two images: the noisy image with noise level $\sigma$ on the left, and a colormap on the right. The colormap shows the contribution, determined by $N_\theta$, of all other pixels in denoising the central red pixel. When the noise is zero, the contributions from all pixels other than the red pixel are zero. As the noise increases, the neural network $N_\theta$ learns to assign weight only to regions that are similar to the red pixel.
>
> ### (C3) Warm-up initialization: what is $x_0$ in Alg. 2?
>
> In all experiments, we initialize with the observed measurement mapped to image space, which is the standard choice:
> - deblurring: $x_0$ is the blurry image,
> - super-resolution: $x_0$ is the bicubic upsampled observation.
>
>
> ### (C4) Algorithm 1: $N_\theta$ takes 3 inputs but (7) shows 2; what is $\sigma$?
>
> Yes, this is a notation issue. Here, $\sigma$ denotes the noise level. When passed to the neural network as an additional constant channel, it acts as a hyperparameter and makes the denoiser noise-aware. Since we use this noise-aware implementation in Alg. 1, we explicitly include $\sigma$ there. However, $\sigma$ does not play any role in the abstract denoiser construction itself: the denoiser can be parameterized using any architecture, with or without a noise channel. For this reason, we did not include $\sigma$ in the general definition in Eq. 7. Writing $N_\theta(\xi,\pi\cdot\xi;\sigma)$ in Alg. 1 would have been a clearer notation, and we will revise this in the final version.
>
>
> ### (C5) Tables 1–2 analysis
>
> Thank you for highlighting this. We agree that the discussion of Tables 1–2 can be streamlined into a single clearer passage, and that the phrasing about ranking can be made more precise to better match the table. We will revise this carefully and proofread the final version for consistency and clarity.
>
> ### (C6) After (9): what is $e$?
>
> $e$ denotes the all-ones image (constant signal). It first appears after Eq. 9 and is used again in Eqs. 15 and 16.
>
> ## (C7) Complexity/runtime
>
> Our denoiser uses a lightweight CNN with only $0.9$M parameters. The computation time scales with the number of permutations $|\mathcal{G}|$ due to the aggregation step. In addition, the symmetrization step $\varphi$ in Eq. 18 increases the cost: a direct implementation can be about $3\times$ slower. In practice, this overhead can be reduced using a cached implementation, and the ICC strategy (Alg. 1) further cuts the loop length by roughly half, leading to a noticeable speedup. We report the computation times below on an NVIDIA GeForce RTX 3090 (49 GB).
>
> |Method|Avg. time per iteration (s)|
> |-|-|
> |DRUNet|0.08|
> |Ours ($D$)|0.74|
> |Ours ($D_{sym}$) (Naive)|2.26|
> |Ours ($D_{sym}$) (optimized)|1.06|
>
> ---
>
> **Summary**
> - We provided a clearer explanation of Fig. 1 to better illustrate what the neural network learns.
> - We clarified that the noise channel is an architectural choice. The abstract denoiser can be parameterized using any neural network architecture.
> - We reported both the computational time and the number of parameters.

---

> > ### Author Rebuttal · Reviewer_6uJm · 2026-04-03
> >
> > First, I would like to thank the authors for their answers that give significant insights.
> > I will increase my score accordingly to weak accept.

---

> > > ### Author Response · Authors · 2026-04-04
> > >
> > > Thank you very much for the follow-up and for increasing your score. We sincerely appreciate your careful review and the constructive feedback, which help us clarify the presentation and evaluation. We are grateful for your time and consideration.

---

### Official Review · Reviewer_mJ1X · 2026-03-11

**Soundness:** 3
**Presentation:** 2
**Significance:** 3
**Originality:** 3
**Overall Recommendation:** 4
**Confidence:** 4

**Summary:**

This paper introduces a trainable, non-expansive denoising architecture designed to guarantee convergence in Plug-and-Play (PnP) reconstruction frameworks. By parameterizing the denoiser as a weighted aggregation of group-permuted inputs and applying a symmetrization technique, the authors ensure the operator maintains a Lipschitz constant of one.  The paper provides a theoretical analysis of the proposed construction and demonstrates its effectiveness through experiments on image reconstruction tasks.

**Compliance With Llm Reviewing Policy:**

Affirmed.

**Final Justification:**

I thank the authors for their rebuttal. The proposed method is interesting, though it has some limitations, such as limited expressiveness and a gap between the theoretical analysis (e.g., the assumption that ξ is fixed) and the practical implementation. Overall, I will maintain my current score.

**Key Questions For Authors:**

1. The theoretical guarantees rely on the symmetrized operator used in Algorithm 1. Could the authors clarify how much the symmetrization step changes the learned denoiser in practice? For example, is there a noticeable difference between the outputs of the original operator and the symmetrized one?
2. The framework relies on a set of permutations to construct the aggregation operator. Could the authors comment on how sensitive the performance is to the choice of the permutation group?
3. The proposed denoiser depends on a reference image ξ obtained from a warm-up reconstruction. Could the authors comment on how sensitive the method is to the choice of ξ, and whether updating ξ during the reconstruction process would improve performance?

**Limitations:**

Yes

**Strengths And Weaknesses:**

Strengths: The paper provides a simple, interpretable construction of non-expansive trainable denoisers to stabilize PnP reconstruction algorithms. The method is theoretically sound and demonstrates competitive empirical performance across benchmarks.

Weaknesses: The proposed denoiser outputs a linear combination of permuted inputs. This may limit its expressive power compared with nonlinear CNN-based denoisers (e.g., DRUNet), which can create new features instead of only averaging existing ones. This gap also appears in the numerical results. In addition, the symmetrization step may further limit the model capacity.

---

> ### Author Rebuttal · Authors · 2026-03-29
>
> Thank you for the positive assessment and for the thoughtful questions. We address the main points below and will be happy to add the discussions in the final paper.
>
> ## (B1) Reply to the Weaknesses
>
> We agree that for a fixed reference $\xi$, our denoiser is linear in the image $x$ during aggregation. This is a deliberate choice that enables the required constraints needed for the robust convergence guarantee (Theorem 3.1) as we do not impose any constraints on the neural network architecture.
>
> At the same time, the method remains expressive because the weights are nonlinear in $\xi$ through $N_\theta$, and the warm-up stage in Algorithm 2 improves the reference used for weight prediction.
>
> We think that introducing controlled nonlinearity in $x$ is an interesting direction, and we will explore it in future work.
>
> ## (B2) Symmetrization step effect on the learned denoiser
>
>  We use two forms of symmetrization to obtain the theoretical guarantees.
> 1. **Inverse-Consistent Condition (ICC)** (Lemma 3.3).
>    This is enforced **during training**, so the network learns under this constraint from the beginning. Empirically, we did not observe any performance degradation due to ICC.
> 2. **The mapping $\\varphi$** (Eq. 18), used to construct $D_{sym}$ with $||D_{sym}||_2 = 1$ (Lemma 3.5).
>    Since $\\varphi$ is applied **after training**, it can introduce a change in denoising. However, this step is essential to obtain a reconstruction operator with provable convergence. Importantly, in practice this **symmetrization does not degrade reconstruction quality**; rather, it improves reconstruction relative to the original non-symmetric, non-convergent system.
>
> For example, in a representative deblurring experiment on CBSD10, symmetrization improves PSNR from **27.39 dB** to **27.70 dB**, while remaining very close to **DRUNet (27.77 dB)**.
> || Unsymmetrized ($D$) | Symmetrized ($D_{sym}$) | DRUNet |
> |-|-|-|-|
> | PSNR (dB)    | 27.39 | 27.70 | 27.77 |
>
>
> ## (B3) Sensitivity to the choice of permutations $G$
>
> **The performance of the method does depend on the choice of the permutation set $G$**. At one extreme, if $G$ is trivial, then the denoiser reduces to the identity operator and provides no denoising.
>
> To study this dependence, we evaluated several representative choices of $G$, including dihedral symmetries ($D_4$), local translations ($T[R]$), and diagonal translations ($DT[R]$) with radius $R$. The denoising results are shown below:
> |Noise Level ($\\sigma$)|$T[7]$|$DT[32]$|$D_4$|
> |-|-|-|-|
> |15|33.64|32.89|29.93|
> |20|32.14|31.37|28.26|
> |25|31.02|30.22|27.01|
>
> These results show that the choice of $G$ has a clear empirical impact. At the same time, the permutation set must satisfy structural conditions such as transitivity and the absence of involutions (Eq. 21) for the theoretical guarantees.
>
> Since $T[7]$ gives better denoising performance while also satisfying the theoretical requirements, we use $T[7]$ in all reconstruction experiments.
>
> ## (B4) Sensitivity to the reference $\xi$
>
> The reference $\xi$ controls the weights through nonlinearity. If $\xi$ is poor (e.g., the initialization), the learned weights can also be poor. This motivates the warm-up stage where we update $\xi_k \leftarrow x_k$ for a few iterations, and then freeze $\xi$.
>
> Freezing $\xi$ is also important from the theoretical viewpoint. The convergence analysis in the paper is established for a **fixed** operator $T_\xi$. If $\xi$ were updated throughout the reconstruction, the operator itself would change across iterations, and the fixed-point contraction guarantee would no longer apply. Thus, the warm-up stage provides a practical way to improve the reference while preserving the theoretical setting needed for convergence.
>
> **The method is not very sensitive to the exact warm-up length**. Fig. 4 already shows stable convergence across several choices. The additional results below show a similar trend: a warm-up of around 20 iterations is a good default, and performance remains stable over a fairly broad range.
>
> At the same time, the warm-up should not be made too long, since excessively delaying the frozen-reference phase can slightly reduce the final reconstruction quality. The second column reports the PSNR of the reference image at the end of warm-up, after which we run 100 iterations with the reference frozen.
>
> |warmup iterations|psnr after warmup|psnr after 100 iterations|
> |-|-|-|
> |0|19.44|24.51|
> |5|25.43|26.57|
> |10|26.53|26.95|
> |15|26.90|27.08|
> |20|27.05|27.13|
> |25|27.11|27.14|
> |30|27.13|27.13|
> |50|27.10|27.11|
> |100|26.84|26.79|
>
>
> We sincerely appreciate your consideration and look forward to your final evaluation.
>
> ---
> **Summary**
> - Symmetrization leads to improved reconstruction performance.
> - The choice of permutations plays an important role in achieving strong results.
> - Warm-up iterations are necessary, but the method is not highly sensitive to the exact warm-up length.

---

> > ### Author Rebuttal · Reviewer_mJ1X · 2026-04-03
> >
> > I thank the authors for the detailed explanation. I believe the method has merit, though it also has limitations (e.g., limited expressiveness and a gap between the theoretical analysis and the actual implementation). Therefore, I will maintain my current score.

---

> > > ### Author Response · Authors · 2026-04-04
> > >
> > > Thank you again for the follow-up and for taking the time to carefully review our responses. We appreciate the confirmation that the concerns are resolved.
> > >
> > > On the remaining point (“gap between theoretical analysis and actual implementation”), we would like to clarify that **our implementation is consistent with the theory presented in the paper**.
> > >
> > > Our convergence guarantee (via contractivity) is stated for the frozen reference image with the *symmetrized denoiser* $\varphi(D)=D_{\mathrm{sym}}$ (Eq. 18). Algorithm 2 implements exactly this setting in all reconstruction experiments. Therefore, the reconstruction results reported in the paper are obtained with the theoretically consistent operator, and the corresponding iterates are guaranteed to converge.

---

### Official Review · Reviewer_Dkvb · 2026-03-13

**Soundness:** 3
**Presentation:** 2
**Significance:** 2
**Originality:** 3
**Overall Recommendation:** 4
**Confidence:** 4

**Summary:**

This work, "Trainable Nonexpansive Denoisers for Contractive Image Reconstruction," proposes a constraint (design) on a neural architecture with 1-Lipschitz by exploiting the action of permutations on the image lattice for the denoisers that are nonexpansive by design. Then, this new denoiser was integrated into image inverse problems such as super resolution and deblurring, proving that the reconstruction operator is contractive under mild conditions. Experiments showed that the proposed method yielded reasonable performance over prior arts, but not the best performance.

**Compliance With Llm Reviewing Policy:**

Affirmed.

**Final Justification:**

All of my concerns were well-addressed in the rebuttal, so I will increase my score from 3 to 4.

**Key Questions For Authors:**

1) I think it is important to clarify the contribution and novelty over prior arts that have the similar / same goals. See below.
- Cohen et al., NeurIPS 2021 was cited, but not properly discussed even though it seems related to this work. Since then, there were a number of works to enhance it such as (T Hong et al., Convergent Complex Quasi-Newton Proximal Methods for Gradient-Driven Denoisers in Compressed Sensing MRI Reconstruction, IEEE Transactions on Computational Imaging 2025).
- There are other works that are similar to the work of Goujon et al., such as (S Ducotterd et al., Improving Lipschitz-Constrained Neural Networks by Learning Activation Functions, JMLR 2024).
- S Zhang et al., Rethinking Gradient Step Denoiser: Towards Truly Pseudo-Contractive Operator, NeurIPS 2025
- M Yukawa et al., Monotone Lipschitz-Gradient Denoiser: Explainability of Operator Regularization Approaches Free From Lipschitz Constant Control, IEEE Transactions on Signal Processing 2025
- U Tanielian et al., Approximating Lipschitz continuous functions with GroupSort neural networks, AISTATS 2021
2) Was the proposed method too restrictive in terms of performance? How about comparing with other recent training based deep learning methods? What are the advantages of the proposed work over them, especially some of the works using large generative models?
3) Can this work be extended to other imaging problems, considering the conditions in Theorem 3.1? For example, there are a number of imaging problems such as phase retrieval, lensless imaging, MRI and so on that have quite challenging imaging operators.

**Limitations:**

No.

**Strengths And Weaknesses:**

This work proposes an interesting framework for constructing a parametric family of denoisers that are nonexpansive by design and theoretically shows that the reconstruction operator with it is contractive under mild conditions. While these results may be good contributions, it is unclear if it is indeed novel over some recent works that were not mentioned (or cited, but discussed properly - it will be important to discuss and compare with other works that have the same / similar goals) and if it is in fact practical in terms of performance, as suggested in the abstract, "training neural networks that simultaneously offer strong denoising performance and global Lipschitz guarantees is challenging." Thus, this work may need to clarify its novelty and usefulness by answering questions on recent works and convincing experiments. Moreover, it is also unclear if the conditions in Theorem 3.1 is indeed practical beyond deblurring and super resolution, which are relatively less challenging in imaging over other complicated imaging operators from compressive sensing literature.

---

> ### Author Rebuttal · Authors · 2026-03-30
>
> Thank you for the careful reading and for highlighting the key concerns. We address the main points below. If our responses address your concerns, we would greatly appreciate your reconsideration of the rating.
>
> ## (A1) Novelty relative to prior art with similar goals
>
> We thank you for pointing out several relevant papers. Our main distinction from them lies in how Lipschitz control is achieved and how the neural network enters the denoiser construction.
>
> **Gradient-step / potential-based denoisers.**
> Works such as Cohen et al. 21 and follow-ups (e.g., Hong et al. 25; Zhang et al. 25) construct denoisers from a scalar-valued neural network, such as an ICNN. The denoiser is then obtained via the gradient of this network. Thus, the denoiser is parameterized by a scalar-valued neural network.
>
> **Lipschitz-network / layer-normalization approaches.**
> Other works (e.g., Ducotterd et al. 24; Tanielian et al. 21) directly enforce nonexpansivity or Lipschitz control at the network level using tools such as spectral normalization, orthogonalization, or Lipschitz-constrained activations. Goujon et al. 23 also use spectral normalization at the layer level. These constraints are typically implemented via iterative procedures, such as power iterations, during training and may restrict the function class.
>
> **Our approach is different.**
> We also use a neural network to parameterize the denoiser, but the parameterization is fundamentally different:
>
> - The learned network $N_\theta$ maps a *pair of images* to a weight map (an image-shaped output), rather than producing a scalar potential.
> - The network $N_\theta$ is *largely unconstrained* (beyond a final positivity operation). We do not impose spectral normalization, orthogonalization, GroupSort, or Jacobian penalties.
> - The Lipschitz control is enforced outside the network, via the aggregation structure together with algebraic conditions on the permutation set (ICC and transitivity).
>
> In short, we neither rely on gradient-based paradigms nor on layer-wise Lipschitz normalization. Instead, we propose a new paradigm obtained by combining (i) an expressive, largely unconstrained learned network and (ii) a structured aggregation. This is complementary to existing convergent PnP paradigms.
>
> We also report a comparison with Goujon et al. 23 and the 1-Lipschitz network of Ducotterd et al 24 on motion deblurring using the 9 blur kernels from Tab. 1 with noise level $0.03$:
>
> |Method|PSNR|
> |-|-|
> |Ducotterd et al.|26.43|
> |Goujon et al.|27.14|
> |Ours|28.29|
>
> ## (A2) Advantage over other training-based methods
>
> Our paper already compares with training-based PnP baselines relevant to our setting.
>
> If the reviewer is referring to end-to-end reconstruction networks, we note that **the main advantage of PnP over end-to-end methods is modularity.** Once a denoiser is trained, it can be reused across different forward operators without retraining the full model. In contrast, end-to-end methods are typically tied to the degradation model used during training.
>
> For example, a SwinIR model (Liang et al. 21) trained for $2\times$ super-resolution with a bicubic kernel drops sharply when tested with the different Gaussian kernels used in Tab. 2, whereas our PnP method adapts without retraining:
>
> |Noise|SwinIR|Ours|
> |-|-|-|
> |0.03|22.81|26.77|
> |0.05|20.72|25.86|
>
> At the same time, **PnP methods can suffer from instability, and therefore may lack reliability. Our work addresses this** by designing a denoiser that is globally nonexpansive by construction and yields a stable contractive reconstruction operator under mild assumptions.
>
> ## (A3) Practicality of Theorem 3.1 beyond deblurring / SR
>
> Thm 3.1 requires a nonannihilating linear forward model, $Ae \neq 0$, to ensure contractiveness of the reconstruction operator. This condition holds for many linear imaging problems, including lensless imaging and MRI when the DC component is sampled. If it does not hold, our denoiser remains nonexpansive; hence, for convex data fidelities, standard PnP schemes such as FBS/HQS still converge by classical fixed-point theory through operator averaging. Phase retrieval falls outside our assumptions, so our framework does not apply there.
>
> To further assess practicality beyond deblurring and super-resolution, we additionally conducted preliminary MRI experiments following the protocol of Goujon et al. On fastMRI (PD, multi-coil, 15 coils, $M_{acc}=4$, $M_{cf}=0.08$, $\sigma=2\times 10^{-3}$), evaluated on the first 10 images, we obtained:
>
> |Method|PSNR|SSIM|
> |-|-|-|
> |Ducotterd et al.|35.65|0.8750|
> |Goujon et al.|37.26|0.9279|
> |Ours|36.14|0.9179|
> ---
> **Summary**
> - We propose a novel framework for nonexpansive denoisers that is distinct from gradient-step methods and 1-Lipschitz networks.
> - The network is largely unconstrained and outperforms directly normalized 1-Lipschitz models.
> - The theory applies broadly to linear imaging problems, including MRI.
>
> We will be happy to add the discussions in the final paper.

---

> > ### Author Rebuttal · Reviewer_Dkvb · 2026-04-04
> >
> > I appreciate the authors' rebuttal. I see all of my concerns were resolved. I will finalize my score after discussing with other reviewers and ACs, but for now, I will be happy to increase my score +1.

---

> > > ### Author Response · Authors · 2026-04-06
> > >
> > > Thank you for the follow-up and for increasing your score. We sincerely appreciate your careful review and are glad our responses addressed your concerns. Thank you again for your consideration.

---

### Decision · Program_Chairs · 2026-04-30

**Decision:**

Accept (regular)

**Comment:**

All reviewers agreed that this is a strong paper with an interesting approach for image reconstruction and should be accepted for publication in ICML